# Inline Raman Spectroscopy Provides Versatile Molecular Monitoring for Platelet Extracellular Vesicle Purification with Anion-Exchange Chromatography

**DOI:** 10.3390/ijms25158130

**Published:** 2024-07-25

**Authors:** Heikki Saari, Heli Marttila, Minna M. Poranen, Hanna M. Oksanen, Jacopo Zini, Saara Laitinen

**Affiliations:** 1Finnish Red Cross, Blood Service, Härkälenkki 13, 01730 Vantaa, Finland; 2Division of Pharmaceutical Biosciences, Faculty of Pharmacy, University of Helsinki, Viikinkaari 5, 00790 Helsinki, Finland; 3Molecular and Integrative Biosciences Research Programme, Faculty of Biological and Environmental Sciences, University of Helsinki, Viikinkaari 9, 00790 Helsinki, Finland; 4Timegate Instruments Ltd., Tutkijantie 7, 90590 Oulu, Finland

**Keywords:** extracellular vesicles, Raman spectroscopy, inline analytics, ion-exchange chromatography, platelets

## Abstract

Extracellular vesicles (EVs) are relatively recently discovered biological nanoparticles that mediate intercellular communication. The development of new methods for the isolation and characterization of EVs is crucial to support further studies on these small and structurally heterogenous vesicles. New scalable production methods are also needed to meet the needs of future therapeutic applications. A reliable inline detection method for the EV manufacturing process is needed to ensure reproducibility and to identify any possible variations in real time. Here, we demonstrate the use of an inline Raman detector in conjunction with anion exchange chromatography for the isolation of EVs from human platelets. Anion-exchange chromatography can be easily coupled with multiple inline detectors and provides an alternative to size-based methods for separating EVs from similar-sized impurities, such as lipoprotein particles. Raman spectroscopy enabled us to identify functional groups in EV samples and trace EVs and impurities in different stages of the process. Our results show a notable separation of impurities from the EVs during anion-exchange chromatography and demonstrate the power of inline Raman spectroscopy. Compared to conventional EV analysis methods, the inline Raman approach does not require hands-on work and can provide detailed, real-time information about the sample and the purification process.

## 1. Introduction

Extracellular vesicles (EVs) are submicron to several micron-small sized membrane vesicles that are gaining increasing interest as mediators of intercellular communication and carriers of therapeutic agents [1,2]. EVs function as carriers of biomolecular information, including for proteins, lipids, nucleic acids, and metabolites and, therefore, can affect the recipient cells in various ways [3]. Current development efforts are focused on exploiting EVs as drug carriers [1] or as biomarkers, for example, in cancer diagnostics [4]. Large-scale isolation methods for EVs are being developed to meet the demand for high-quality EV preparation in large quantities as needed for these novel applications. The current isolation methods include size-based separations, such as tangential-flow filtration, and precipitation and surface interaction-based methods, such as ion-exchange chromatography, which can process even large sample volumes relatively quickly. Ref. [5] Due to the heterogeneous structure and complex composition of EVs and their physicochemical properties (size, density, and surface charge), the analysis of EV samples requires a combination of many different approaches, including particle size analysis (for example, Nanoparticle Tracking Analysis, NTA), the enrichment of EV markers (Western blotting), and morphology analyses (electron microscopy) that are recommended by the International Society of Extracellular Vesicles. Ref. [6] These analyses are usually performed downstream from isolation and, taken together, can be very laborious and time consuming. Inline monitoring of EV purification is therefore necessary, not only to save time and costs but also to obtain a more detailed, real-time view of the process.

Inline detectors are most convenient for methods using a liquid flow, such as liquid chromatography, where UV detection at 280 nm (A280) is most commonly used to monitor protein signals. However, since EVs may not contain many proteins per vesicle, they can remain largely undetected by UV absorbance. Nevertheless, it is useful to follow an A280 signal to monitor the elution of impurities and the overall purification process. Other wavelengths can also be used, but A260, for example, is even less informative, since nucleic acids that are detected at 260 nm are usually even less prominent in EV samples than EV-associated proteins [5,7]. Moreover, by use of multi-wavelength UV detectors, it is possible to monitor these different classes of molecules simultaneously, which is useful for the characterization of the sample composition, for example, the nucleic acid/protein ratio using A260/A280. The whole absorbance spectra can also be monitored with a diode array detector, which can give a much more detailed view of the different molecules in the sample. Ref. [8] In addition to measuring the absorbance, the attenuation of the light-UV region, which is the quantity measured using the UV detectors, also depends on light scattering. This is caused by the turbidity of the sample and has been suggested as an alternative method for identifying EV-containing fractions and even measuring their size. Ref. [9] Turbidity is usually more distinguishable at longer wavelengths (>400 nm), where biological samples typically do not have high absorbance, and therefore do not cause a high background. Ref. [10] Light scattering is one of the most used approaches for analyzing EVs overall due to its sensitivity, for example, with NTA [11]. Thus, light-scattering detectors for liquid flow methods may be more specific for EV detection than UV detectors, because they are sensitive to EVs regardless of their composition and do not give as high of a signal for single macromolecules such as proteins. Light scattering can be used to differentiate and identify fractions from preparative chromatography that contain EVs and even estimate their concentration and size [5,12].

Using fluorescent tags independently or in conjunction with light-scattering detectors can further increase the accuracy of identifying EV-containing fractions or quantifying EVs using size-exclusion chromatography (SEC), a method termed “Flu-SEC”. Ref. [13] Due to the requisite of labeling, such an approach is limited to analytical approaches, unless the desired EV product contains a fluorophore. In such cases, fluorescence detection could be part of preparative chromatography monitoring as well. Coupling multiple types of analytics, such as UV, fluorescence, and light-scattering detectors to a size-based separation technique such as field-flow fractionation (FFF) is a powerful analytical technique for determining the size distribution of an EV sample [5,14]. Given the complex, heterogeneous nature of EVs, such multi-detector setups are most likely the best approach to monitor the purification process as well. However, these measurements may need to account for interference effects; since EVs and other nanoparticles can cause significant background signals and vice versa, fluorescence can be detected as background by light-scattering detectors [15].

In the present study, we demonstrate the utility of using an inline Raman detector to monitor the purification of platelet EVs using monolith anion-exchange chromatography, as shown in Figure 1. Platelets from donated blood were washed and incubated overnight to produce EVs. The collected platelet supernatant (PS) was used as the starting material for chromatographic EV purification. The collected flowthrough (FT) and eluate from the column were collected and analyzed with conventional EV characterization methods to identify EV-containing fractions and evaluate their purity. An inline Raman detector was coupled to the chromatography system and was used to collect Raman spectra along the system’s own UV detector during the EV purifications. Raman spectroscopy allows for the direct analysis of the functional groups in a sample due to their vibrational modes, which cause photons to scatter at characteristic wavelengths, making it a potential method for assessing EV purity [16,17]. Since several types of biomolecules can be identified simultaneously in a concentration-dependent and non-destructive manner that can also be enhanced with surface engineering of the detector surface, Raman spectroscopy has exceptional potential to monitor EVs and other biological products with a complex chemical composition [18,19]. Anion-exchange chromatography is a well-suited method for coupling with inline Raman monitoring, since the process simultaneously concentrates and purifies EVs, and it can be operated in a closed system through all the steps. During anion-exchange chromatography, negatively charged analytes, such as EVs, bind to the positively charged ligands of the anion-exchange column via ionic interactions. This allows for the removal of impurities that do not possess a sufficient negative charge, followed by the elution of EVs with an increased salt concentration [5,20,21,22,23,24]. Here, we used the inline detector to measure Raman spectra at set intervals during the sample injection, washing of the unbound material, and elution. With this setup, it was possible to distinguish between these phases based on their molecular compositions. A further downstream analysis of the purification process using Western blotting and NTA revealed a good separation of EVs from impurities, such as free proteins and Very Low- or Low-Density Lipoprotein (V/LDL) particles from residual plasma. Additionally, the Raman spectra suggest that the platelet EVs contain carotenoids, which have been considered to being carried only by LDL particles in blood.

## 2. Results

### 2.1. Anion-Exchange Chromatography Effectively Captures EVs from the Platelet Supernatant

As the model isolation method, we used anion-exchange chromatography, which was challenged using varying sample types to assess how the sample composition and amount affect the chromatographic purification of EVs and if any differences could be identified in the elution profiles with an inline Raman detector. We used three types of EV-containing PS samples: (1) standard load (SL), aimed for a load well within the column’s capacity, (2) high load (HL, double the SL), aimed for a higher yield of EVs and potentially partial loss in the FT, and (3) SL in 1% (*w*/*v*) bovine serum albumin (BSA) (SB), for the evaluation of the effect of an additional impurity. The separations were monitored with the UV detector of the chromatography system at 280 nm (for representative chromatograms, see Figure 2A). Since the HL and SB samples were twice the volume of the SL, their FT peaks were wider as well. During the elution step, a rise at A280 was observed immediately when conductivity began to increase at the end of fraction 2 (Fr2), with most of the elution peak collected in fraction 3 (Fr3).

To monitor the presence of EVs, we performed a Western blot analysis of the collected fractions for EV markers, tetraspanins CD9 and CD63 (Figure 2B–D), and NTA for particle distributions (Figure 2E). Additionally, we analyzed the distribution of ApoB-proteins (ApoB-100 and ApoB-48) that are present in lipoprotein particles (Figure 2D). Lipoproteins are common in blood-derived samples and have physicochemical characteristics similar to EVs, which makes them one of the most problematic impurities in the blood-derived EV samples. The distributions of EV marker proteins CD9 and CD63 followed a distinct pattern in the FT and elution fractions, with low variation between individual experiments. The highest marker intensities were observed in the peak fraction Fr3. Especially, CD9 was enriched in Fr3, and only low amounts of the protein were detected in FT and other fractions. While CD63 followed a similar pattern, it was observed in the FT at a higher intensity than CD9. The smear-like migration pattern of CD63, as well as its distorted shape in the FT fraction, caused by a high number of other proteins in the 37–50 kDa range, could partially explain this observation, making the quantitation of CD63 less reliable than that of the distinct single band of CD9 (Figure 2D). Overall, Western blotting can be considered a semi-quantitative method due to its complexity, but here, the results indicate a clear pattern. Regarding the different sample types (SL, HL, and SB), only minor differences were found. With the HL sample, there appeared to be slightly more CD9 in the FT, but this was not statistically significant. Counterintuitively, the SB samples resulted in slightly higher EV marker enrichment in the elution Fr3, which was also found to be statistically significant. Since BSA is a negatively charged protein under the separation conditions, it would be expected to compete for binding to the column with the EVs, leading to a reduced EV yield in the elution. Instead, we found that the monolithic column bound almost no BSA, even when using BSA alone without the PS at the same separation conditions. Therefore, it would not compete for column binding with the EVs either, explaining our observations.

Of the ApoB proteins, only the ApoB-100 isoform was detected in most samples, indicating the presence of V/LDL particles (Figure 2D). The isoform ApoB-48, a chylomicron-bound apolipoprotein, was not detected, designating the absence of chylomicrons. Chylomicrons are likely removed from the samples more efficiently during platelet preparation through centrifugation, as they have a very low density and float to the top quickly. The ApoB-100 protein was, in contrast to the EV markers, enriched in the FT, being only barely detectable or not at all in the elution fractions. These results suggest a good separation of V/LDL particles and EVs during the process. As EVs in plasma have been previously described as being partially associated with these lipoproteins or containing the ApoB-proteins themselves [24], it is also possible that the small amount of ApoB detected here is also EV associated instead of being associated with free lipoprotein particles. Determining the distribution of nanoparticles showed that there were more particles in the FT than in Fr3, at a ratio of about 2:1. While nanoparticles detected using NTA in scatter mode cannot be distinguished as EVs or any other particles, the results still indicate that most particles in the sample bypass and, thus, are less negatively charged than the ones bound and eluted from column. Considering the EV and lipoprotein marker distributions between the FT and Fr3 (Figure 2B–D), it seems that while the elution fractions contained EVs, the FT contained other particles such as lipoproteins and some EVs.

### 2.2. EVs Are Enriched in the Eluate and Purified during Anion-Exchange Chromatography

To compare the overall protein compositions and EV purities of the different fractions and sample types, we performed protein load-normalized SDS-PAGE and Western blot analysis, analyzing the elution peak Fr3 together with the FT and starting material PS (Figure 3). Each sample type had its distinct protein patterns (Figure 3A). The FT samples closely resembled the PS with the exception of the SB sample FT, which contained a very prominent band corresponding to BSA. The protein patterns of the Fr3 samples were completely different from those of PS and FT, as they lacked most of the intense bands of PS and FT and had several bands enriched, which were not visible in the PS and FT samples. The EV marker proteins CD9 and CD63 were significantly enriched in all Fr3 samples compared to PS samples and were reduced or completely absent in FT (Figure 3B,C). While there was no statistically significant difference between the different sample types of Fr3, the HL sample FT had higher EV marker enrichment than SL or SB sample FTs, which suggests a higher loss of EVs into the FT. ApoB, in contrast, was abundant in all FT samples and low in all Fr3 sample types (Figure 3D).

Comparing the physicochemical properties of the nanoparticles in the PS starting material, FT, Fr3, and high-purity reference EVs (REVs) revealed clear differences between the different sample types. Regarding size, the FT was found to contain the smallest median nanoparticles, with a size distribution more inclined towards particles with diameters of 50–100 nm than other samples, with a median diameter of 97 nm (Figure 4A–C). Both anion-exchange purified nanoparticles and REVs were larger than those in the FT, with median diameters of 130 and 122 nm, respectively. The diameters of PS nanoparticles were between these groups. It should be noted that the detection limit of NTA causes particles smaller than 50 nm in diameter to be essentially undetectable in our samples [11]. For this reason, their presence cannot be evaluated based on these results. Another physical characteristic of nanoparticles we analyzed is the zeta potential, which is related to the surface charge. EVs have been described as having a strongly negative zeta potential [25], which makes anion-exchange chromatography a suitable method for their purification, as it binds negatively charged particles. The zeta potential results are presented in Figure 4D–F. The zeta potential distribution of PS showed high variability between individual samples, most likely due to its heterogeneous composition, which may not be suitable for an accurate zeta potential determination. The nanoparticles in Fr3 and REVs both had a similar negative zeta potential with a mean value of about −40 mV. In comparison, the particles in the FT were much less negatively charged, with mean zeta potentials from about −10 to −5 mV. This result is in line with the separation principle of anion-exchange chromatography, which should favor the binding of the most negative particles to the column, allowing other less negative particles to be excluded in the FT.

While the SL Fr3 samples had many properties comparable to the REVs, they were not as pure. All Fr3 samples contained more protein bands in the SDS-PAGE analysis, and their EV markers were not as enriched as in REVs (Figure 5A,B). The protein composition of the Fr3 samples was also much more variable than that of REVs, which were nearly identical in composition across all six biological replicates. ApoB staining also showed faint bands in Fr3 samples but not in REVs. This was expected, since anion exchange alone cannot separate EVs from all anionic proteins, although good specificity towards EVs was observed here. On the other hand, the REVs were prepared using multimode size-exclusion chromatography, which adsorbs proteins < 700 kDa, combined with ultracentrifugation, which separates V/LDL lipoproteins from EVs due to their difference in density. Based on the biochemical and physical characterizations, we found no significant differences between the two different approaches for preparing the REVs: with a preconcentration step or without prior to the Capto Core 700 purification. CryoTEM imaging of REVs confirmed the presence of small vesicles about 50 to 400 nm in diameter and various degrees of lamellarity but no lipoproteins or other particles (Figure 5C). In conclusion, during anion-exchange chromatography, EVs and some of the anionic impurities present in PS are enriched in the eluate, are relatively large in size, and have a strongly negative zeta potential compared to other nanoparticles in the FT, which includes lipoproteins and possibly other nanoparticles with a smaller size and less negative zeta potential.

### 2.3. Inline Raman Spectroscopy Can Identify Separate Purification Phases and Their Deviations

In contrast to the plethora of manual techniques for the downstream monitoring of the chromatographic fractions presented in the previous sections, real-time monitoring with the inline detectors did not require any hands-on work. The UV chromatograms show two main peaks: the FT and the elution peak associated to the EVs, interspersed by a washing phase (Figure 2A). FT UV profiles show some differences in their peak size and shape between the three sample types, which are caused by differences in sample injection. The biochemical nature of these differences is not disclosed by the UV data, which only indicate what sample volume and approximate protein load is injected and eluted. Time-gated Raman data, on the other hand, showed remarkable differences between the chromatography fractions (Figure 6). The Raman data clearly revealed a difference in the SB FT compared to the other two sample types (Figure 6B–D). Due to the added BSA, a clear increase in protein-associated peak intensity was observed (notably the 750, 1060, 1480, and 1660 cm^−1^ peaks), which made the other peaks appear relatively smaller, explaining the difference. Some of the most notable peaks are listed in Table 1.

While there were similar Raman peaks in both FT and eluate fractions, indicating the presence of shared biomolecules, their relative intensities allow for assigning the fractions into distinct phases. Statistical analysis of such complex data provided a detailed insight on the different runs and their phases (Figure 7). Figure 7A shows the Raman spectra measured using the SL sample type, with the most significant Raman regions identified according to their associated biomolecular species. Principal component analysis (PCA) on time-gated Raman spectra of the measured points during the separation clustered them according to the separate phases of the SL chromatogram (Figure 7B). While the PC1 component had the highest score, the elution fractions were separated from the rest by PC2 (Figure 7C). The FT cluster is dispersed alongside PC1 due to the long tail of the fraction and sample dilution, due to the washing step. Nonetheless, the peak FT fractions were separated from the other phases based on PC1, which described the vast majority of the variance (98.7%), while PC2 was only 0.27%. This is because PC1 is associated with the presence of carotenoids (Figure 7C, peaks at 1008, 1157, and 1518cm^−1^), which gave a very strong signal, while PC2 can be partially associated to C-H vibration (1060 cm^−1^) and Amide I (1660 cm^−1^), which are ubiquitous to lipids and proteins and, therefore, not as clearly enriched in any fraction.

These results provide further validation that the separate phases of the chromatography-based EV purification process can be identified based on differences in the molecular components visible in the Raman spectra. We examined whether the spectral analysis could identify differences caused by the different sample types (SL, HL, and BSA) in any phase (Figure 7D,E). In the FT fractions (Figure 7D), the BSA samples formed their own cluster, which could be clearly separated from the others. This can be explained by the obvious difference in composition, caused by the added BSA. In all sample types, the spectra collected from the elution fractions (EVs) all cluster in such an overlapping pattern that they cannot be distinguished from each other (Figure 7E), suggesting that there are no significant differences in their compositions. In other words, the EV fractions eluted were very similar regardless of the sample type. It should be noted that the positioning of individual spectra in the PCA plot is also related to their collection phase during the chromatographic separation. This is because unlike the peak fractions, the spectra of the fractions with a low concentration are more difficult to separate from each other regardless of sample type, as they more closely resemble the buffer.

Partial Least Squares Discriminant Analysis (PLS-DA), a multivariate inverse least squares discrimination method, was used to classify the fractions with three sample types (SL, HL, and BSA) and possibly identify whether, for example, the column was not functioning as expected, leading to the poor separation that is reflected in the Raman spectra (Figure 8). In each PLS-DA, the Raman spectra collected with SL sample was used as a training set for the model (Figure 8A(I–III)), and SL (Figure 8B(I–III)), HL (Figure 8C(I–III)), or SB (Figure 8D(I–III)) were used as the test sets. The results indicate that the PLS-DA based model is efficient in predicting the EV, FT, and buffer fractions in other standard runs. Based on the analysis, a high sensitivity and specificity were predicted for the EV fraction and lower values for the FT and buffer (Table 2). This is because the FT has a long tail, where the Raman signature fades out slowly. Based on PLS-DA model, the FT of the HL sample also contained EVs (Figure 8C(I)). Similarly, the experimental data (Figure 2B,C and Figure 3C) also suggested that EVs may be leaking into the FT when a higher sample load is used (HL sample). Based on these observations, it might be advised to switch to using a larger column or optimizing the separation conditions to minimize EV losses into the FT. Taken together, the results obtained from analyzing the collected Raman spectra were well in line with the biochemical and physical assays performed downstream: (1) The FT fractions had a significantly different composition from the elution fractions, caused by the separation of EVs and other anionic components. (2) The elution fractions were similar regardless of the loaded sample volume (SL vs. HL), (3) The added impurity (BSA) was identified as a deviation in the FT of the respective runs.

## 3. Discussion

While there is increasing interest in investigating EVs as versatile therapeutic agents and conveyors of intracellular communication, there is a clear need for monitoring their production and isolation processes. EVs can be produced with virtually any desired cell type in high quantities with bioreactors and purified with size- and/or surface interaction-based strategies, which are also being developed continuously. Also, the donated platelets that we used in our study are a by-product that are not in as much demand in transfusions as erythrocytes, but platelets may serve as a source for therapeutic EVs [29,30]. As such, donated platelets are often stored with blood plasma, which adds another challenge to the production of platelet EVs, besides scalability and cost efficiency. Along with efficient EV purification methods, new analytical approaches are also needed for process monitoring if EVs are ever to be produced on a commercial scale as well as to improve the cost efficiency of EV research and development. For this purpose, we suggest inline Raman as a novel method to identify and assess EV-containing fractions in liquid flow-based methods. Due to the complex, heterogeneous structure and composition of EVs, several analytical methods are typically needed to assess variations in EV quality. The commonly used UV-based detection at 280 nm, for example, cannot distinguish between peaks containing EVs or impurities, limiting its usefulness for process monitoring and development. This is, of course, the limit of using a single wavelength setup targeted at protein detection, and more wavelengths could be used for more informative monitoring. However, UV spectroscopy is still not very specific in general, as there is much overlap between the absorption spectra of different biomolecules. Raman spectroscopy is particularly useful in this regard, since it can be used to assess several distinct types of biomolecules at once, as has also been demonstrated in the case of EVs [16,17], and it is relatively insensitive to water background, unlike its “sibling method” infrared spectroscopy. EVs are, for example, often the main source of lipids from a given source, and lipids can be identified in the Raman spectrum and, therefore, be used to identify the EV-containing fractions.

With blood-derived samples, EV isolation and analysis is still not simple, as they contain a complex mixture of different components with lipids, namely lipoproteins. Nevertheless, the anion-exchange chromatography with a QA column was able to separate most of the interfering V/LDL lipoproteins to evaluate the use of an inline Raman detector during platelet EV purification. As the Raman signal in this setup is relatively weak, statistical methods are needed to identify the fractions of interest and any variations therein. Based on the collected spectral analysis, the Raman detector used here was able to differentiate between the separate phases of the purification process, as well as variations in the FTs of the three types of samples. We challenged the separation and detection setup by increasing the sample load and introducing an impurity (BSA). The findings were consistent with our downstream analyses, concluding that neither challenge significantly affected the composition of the eluted EV fraction, even though an increased sample load increased the total eluate yield, and the added BSA was found to mainly alter the FT fraction. Among the identified Raman peaks, we found that the platelet EV fraction contained also carotenoids, which give a particularly strong Raman signal in the ~1000, ~1160, and ~1520 cm^−1^ areas due to their characteristic resonance effect [31]. Normally, lipoprotein particles are considered as the primary carriers of carotenoids in the blood [32], but platelets have also been found to contain carotenoids. Ref. [33] Therefore, it is natural that their secreted EVs would also contain the same carotenoids, although their biological significance has not been clarified yet. Due to their high intensity, the carotenoid peaks were the main cause of variance as identified by the principal component analysis, used for separating the flowthrough from other fractions. The other peaks identified were less specific to any fractions and, thus, did not provide as high of an explanation factor. Besides the carotenoids, the qualitative chemical differences between these materials are subtle, since both contain lipids, proteins, and nucleic acids and, therefore, the same functional groups (and same Raman signature). The only difference is in the relative amounts of the functional groups, i.e., different types of lipids and/or proteins or their combinations in different ratios. Hence, their usefulness in the spectral separation of different phases is more limited. The carotenoid peak could potentially cause problems in analyzing platelet EVs with Raman spectroscopy due to the high intensity of the peak, but the problem can be addressed by using other laser wavelengths that do not cause a resonance effect.

While the EVs in the eluate were not as pure as the reference, EVs derived with a multi-step protocol, the monolith anion exchanger, showed surprisingly good specificity towards EVs, removing protein impurities, including most of the V/LDL particles. Comparable purity in the end product was observed in all three loading conditions, showing that the method was not significantly affected by the sample volume or added impurities. Ion-exchange chromatography has several strengths regarding its application in EV isolation, including an easily scalable sample capacity, concentration of EVs from large volumes, and separation of impurities based on surface structures rather than particle size. The last point is especially relevant for blood-derived samples, as the lipoproteins overwhelmingly outnumber any EVs in blood and, due to their similar particle size, cannot be completely separated with size-based methods only. Besides anion-exchange chromatography, cation exchange can also be used as an alternative method for EV purification for different selectivity [21,34,35]. For further improved purity, size-exclusion chromatography could be added as a polishing step after ion exchange for the size-based separation of impurities from the EVs and buffer exchange to remove excess salt [34,35,36]. This approach may well become a future standard for EV isolation from a variety of sample types and replace ultracentrifugation. Unlike ultracentrifugation or other manual methods, separation techniques that use liquid flow can also be equipped with a plethora of inline detectors for the real-time monitoring of the purification process and EV characterization. Here, a Raman detector may compliment the other detection methods well by providing a good overview of molecular characteristics in a sample, identifying EVs from other nanoparticles and small contaminants in a label-free fashion.

## 4. Materials and Methods

### 4.1. Platelet Supernatant Preparation

Expiring platelet concentrates were acquired from Finnish Red Cross, Blood Service. The preparations were medical-grade platelet concentrates, collected and pooled from the donated blood of four blood-type-matched donors, and stored with plasma. All donated blood products used for research were treated anonymously and were obtained from blood donors who had given their informed consent. The research was in accordance with the rules of the Finnish Supervisory Authority for Welfare and Health (Valvira, Helsinki, Finland). The platelets were collected from the concentrates via sedimentation with a Lynx superspeed centrifuge (Thermo Fisher Scientific, Walton, MA, USA), using the rotor T12 at 4000× *g* for 30 min at +4 °C. The platelets were suspended into 25 mL of 50 mM Tris and 100 mM NaCl at pH 7.5, agitated with a vortex mixer at medium speed for one minute to induce platelet activation, and incubated overnight at room temperature under mild agitation. The cells were removed via centrifugation at 4000× *g* for 30 min at +4 °C, and the platelet supernatant (PS) was collected and filtered with a 0.45 µm pore size PVDF syringe filter (Sartorius, Göttingen, Germany). Using a 25 G needle to collect the supernatant helped prevent the platelets from entering and clogging the filter membrane.

### 4.2. Anion-Exchange Chromatography

The ÄKTA Pure 25 chromatography system (Cytiva, Marlborough, MA, USA) with UV and conductivity detectors and the fraction collector F9-R was used for all chromatography experiments at the Biocomplex Purification Unit at the University of Helsinki. A 1 mL CIM QA large-pore monolith column (BIA separations, Ajdovščina, Slovenia) was used for the anion-exchange experiments with a 5 mL sample loop and a flow rate of 1 mL min^−1^. The binding and washing buffer (buffer A) was 50 mM Tris and 100 mM NaCl at pH 7.5, and the elution buffer (buffer B) was 50 mM Tris and 2 M NaCl at pH 7.5. The column was prepared by washing with 10 mL of 50% buffer B (1 M final NaCl), followed with about 10 mL of 100% buffer A, until the conductivity stabilized. The binding and elution protocol was the following (Table 3):

After each run, the column was cleaned with a solution containing 0.5 M NaOH and 2 M NaCl and prepared again as described above.

### 4.3. Reference EV (REV) Preparation with Multimodal Size-Exclusion Chromatography and Ultracentrifugation

REVs were purified from PS through a multi-step method using differential ultracentrifugation (UC) and multimodal SEC. A 20 cm EconoColumn (Bio-Rad, Hercules, CA, USA) glass column with 1 cm i.d. was packed with Capto Core 700 resin (Cytiva) with a final column volume (CV) of approximately 15 mL, using 0.4 M NaCl in 20% (*v*/*v*) ethanol according to manufacturer instructions and fitted with a 15 cm flow adapter. The column was equilibrated with running buffer (50 mM Tris, 100 mM NaCl, pH 7.5) until the conductivity stabilized. The flow rate was 1 mL min^−1^. Each run started with a 0.5 CV column equilibration step, followed by 10 mL of PS sample injection with a 10 mL sample loop. Fraction collection was started after about 0.5 CV, once the A280 started to rise and was continued for about 30 mL until the A280 dropped and stabilized, resulting in one large fraction.

After each run, the column was cleaned with 3 CV of 30% (*v*/*v*) isopropanol in 1 M NaOH run in the reverse direction, including a 15 min incubation step before rinsing the column with milli-Q water, followed by re-equilibration in running buffer.

The EV-containing, 30 mL Capto Core 700-purified flowthrough was then concentrated using UC, using the ultracentrifuge Sorvall WX Ultraseries (Thermo Fisher Scientific) with the rotor AH629 (100,000× *g* for 1 h at +4 °C). After removing the supernatant, the pellet was resuspended into 100 µL of running buffer.

Alternatively, EVs were first pre-concentrated from 25 mL of PS using UC (as above) and resuspended in running buffer (final volume about 200 µL), followed by Capto Core 700 purification with a 500 µL loop operated as above. Instead of one large fraction, 1 mL fractions were collected from sample injection, with a total of 15 fractions. The peak fractions 6–10 were concentrated with UC (rotor AH650 at 100,000× *g* for 1 h at +4 °C), and the resulting pellet was resuspended with Tris-buffered saline. The REV samples produced using this alternative SEC + UC approach were samples #5 and 6, while samples #1–4 were produced using the UC + SEC + UC approach.

### 4.4. Nanoparticle Tracking Analysis

Nanoparticle Tracking Analysis measurements were performed with the ZetaView (Particle Metrix, Inning am Ammersee, Germany) particle analyzer in scatter mode, equipped with a 488 nm green laser and software version 8.05.16 SP3. The samples were diluted in distilled water for the measurements in a ratio of 1:1000–1:10,000. Sensitivity was set at 85 and the shutter at 100. The size distribution and particle concentrations were measured at 11 positions with one measurement cycle. The zeta potential was measured with the same settings in continuous mode with two stationary layers in two cycles.

### 4.5. SDS-PAGE and Western Blotting

The sample volume was normalized as either an equal volume or protein mass to assess the protein yields or compositions, respectively. The protein concentration was measured at A280 nm with a NanoDrop ND-1000 (NanoDrop Technologies, Wilmington, DE, USA). To equalize the volumes of chromatography fractions, the FT samples were concentrated by 15-fold (the FT total volume was 15 times higher than that of individual elution fractions) with Amicon Ultra-0.5 10 kDa ultrafiltration units (Merck Millipore, Burlington, MA, USA) at 14,000× *g* for 15 min at RT. After concentration, 5 µL of 4 × Laemmli sample buffer (Bio-Rad) was added to the filter unit to collect all vesicle material by lysing them, and the retentate was collected by inverting the filter unit into a clean collection tube and spinning at 1000× *g* for 2 min. Samples were prepared in Laemmli sample buffer under non-denaturing conditions for 10 min at +99 °C and analyzed in a 4–20% Mini-PROTEAN^®^ TGX StainFree™ Protein Gel (Bio-Rad) at 200 V with Strep-tagged Precision Plus Protein™ Unstained Protein Standard (Bio-Rad). The gel was imaged with ChemiDoc Touch (Bio-Rad) imager with a stain-free gel protocol.

Proteins were then transferred from the gel to a PVDF membrane with the Trans-Blot Turbo Transfer system (Bio-Rad) with a Trans-Blot Turbo RTA Mini 0.2 µm PVDF Transfer Kit (Bio-Rad). The proteins were visualized on the membrane by incubating it in methanol for 10 min followed by Ponceau S staining solution (Thermo Fisher Scientific) and cutting the membrane between the 100 and 75 kDa markers. The membrane was blocked for 30 min in 3% (*w*/*v*) BSA in Tris-buffered saline 0.1% (*v*/*v*) Tween20 (TBS-T). After blocking, the membrane with >75 kDa proteins was incubated in anti-ApoB (rabbit polyclonal Anti-Apolipoprotein B antibody (Abcam, Cambridge, UK, cat. #ab20737), 1:1000) in 3% BSA TBS-T and the membrane with lower mass proteins in anti-CD9 (mouse anti-human CD9 (BD Pharmingen, Franklin Lakes, NJ, USA), cat. #555370, 1:1000) + anti-CD63 (mouse anti-human CD63, BD Pharmingen, cat. #556019, 1:1000) overnight at +4 °C. After incubation, the membranes were washed three times in TBS-T and incubated in secondary antibody-HRP (Goat Anti-Mouse IgG (H + L)-HRP Conjugate, Bio-Rad, cat. #1706516, 1:1000 or Goat Anti-Rabbit IgG (H + L)-HRP Conjugate, Bio-Rad cat. #170-6515, 1:1000) + StrepTactinHRP (Precision Protein™ StrepTactinHRP Conjugate, BioRad, 1:5000) in TBS-T for 1 h at RT. The membranes were again washed three times in TBS-T and once in TBS, followed by detection using the Clarity Western ECL Substrate (Bio-Rad) with the ChemiDoc Touch imager chemiluminescence protocol. The acquired images were processed, and band intensities were measured with ImageJ software version 1.54d.

### 4.6. Cryo-Transmission Electron Microscopy

Cryo-transmission electron microscopy (cryoTEM) was performed at the cryo-EM unit at the University of Helsinki, Finland. The REV samples were vitrified on glow-discharged electron microscopy grids, Quantifoil holey carbon R1.2/1.3 Cu 300 mesh (Quantifoil, Grosslobichau, Germany), using a Leica EM GP plunger (Leica microsystems, Wetzlar, Germany) at 80% humidity and 1.5 s blotting time using front blotting.

Cryo-EM grid screening and data collection were performed using a Thermo Fisher Scientific Talos Arctica operating at 200 kV and equipped with a Falcon 3 direct electron detector operating in linear mode. Images were collected at 57,000× magnification, image size of 4096.4096, and at a 0.26 nm/pixel sampling rate.

### 4.7. Raman Measurement

The Timegate PicoRaman M3 (Timegate Instruments Oy, Oulu, Finland) with a 100 ps 532 nm laser and complementary metal oxide single-photon avalanche diode was connected to an ÄKTA Pure 25 chromatography system via ProbeProMini (Timegate Instruments) and SCHOTT ViewCell (Schott, Mainz, Germany). Raman spectra were measured continuously throughout the chromatography run with a 20 s acquisition time, 88 mW as the laser power, and a spot size of 100 µm. The principles of time-gating are introduced in [37].

### 4.8. Statistical Analysis

Statistical analysis was performed using GraphPad Prism software version 9.5.1 (GraphPad, La Jolla, CA, USA), using one- or two-way ANOVA, where indicated, with Fisher’s LSD multicomparison test. Raman data were exported and pre-processed (virtual gated, and smoothed) by TGLab software (Timegate Instruments) and processed in PLS Toolbox (Eigenvector Research, Manson, WA, USA). Figure 6 spectra were baseline subtracted, smoothed, and normalized by the Standard Normal Variate (SNV). For the multivariate analysis, in both PCA and PLS-DA, the spectra were pre-processed by Extended Multiplicative Scatter Correction (EMSC) and mean centered. The cross-validation method was the Venetian blind method in both cases.

PLS-DA was performed, selecting two latent variables, while for PCA, four principal components were selected.

## Figures and Tables

**Figure 1 ijms-25-08130-f001:**
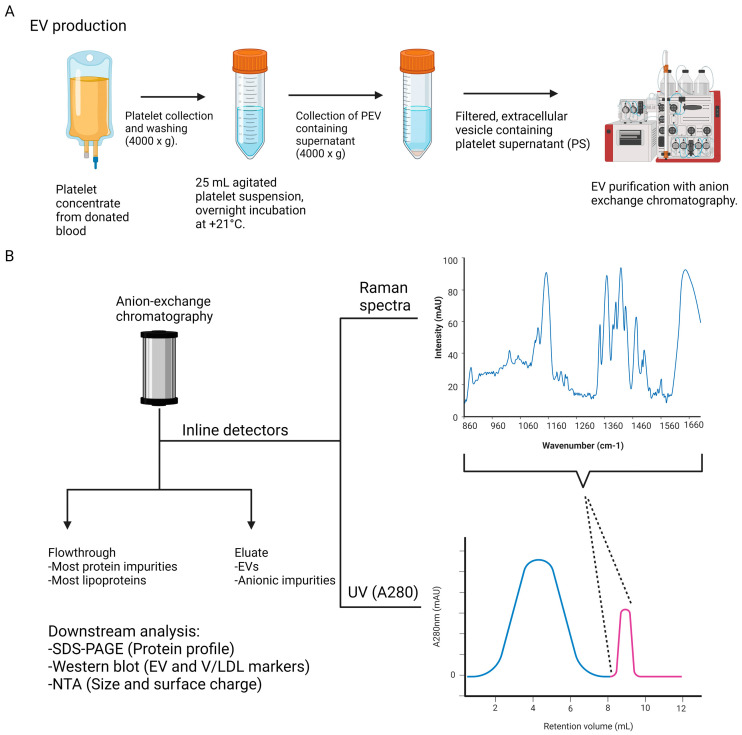
Purification process and monitoring of EV sample purity using anion-exchange chromatography coupled with Raman spectroscopy and other downstream analysis mehods. (**A**) Platelet EVs (pEVs) were prepared from donated platelets and purified using fast protein liquid chromatography with a monolithic anion exchanger. The platelets were washed and collected, followed by activation via agitation and overnight EV secretion in the platelet supernatant (PS). (**B**) The purification process was monitored using the chromatography system’s built-in UV detector measuring at A280 nm, as well as an attached inline Raman detector measuring Raman spectra at set intervals. Additionally, downstream analysis was performed with both flowthrough and eluate fractions for overall protein and nanoparticle characteristics and EV markers. Image generated with Biorender.com.

**Figure 2 ijms-25-08130-f002:**
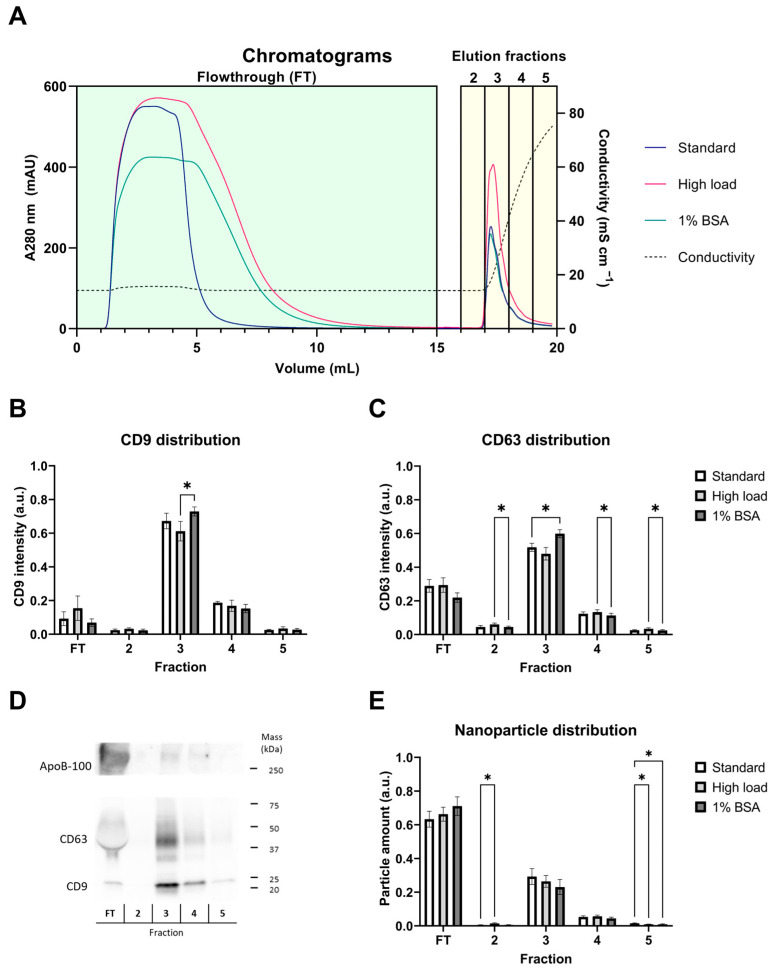
Monolithic anion-exchange chromatography separates EVs from most of the contaminants in platelet supernatant. (**A**) Representative chromatogram of each sample type (standard, high load, and added BSA), monitored with a UV detector at 280 nm. The flowthrough (FT) included sample injection into the column and washing of unbound material. FT was collected into one fraction for downstream analysis. Bound material was eluted with a high-salt buffer, indicated by an increase in conductivity, and 1 mL fractions were collected. Fraction 1 was column void volume and not collected. (**B**,**C**) Assessment of EV marker distribution (CD9 in B and CD63 in (**C**) in the fractions using Western blotting. Equal volume loading was used to normalize the sample volume in the assay, representing the total yields in each fraction. (**D**) A representative blot, including ApoB staining also, where only the ApoB-100 isoform is visible. (**E**) Nanoparticle distribution in the fractions, as assessed using Nanoparticle Tracking Analysis. The results in B, C, and E were normalized against the total sum of marker intensity or particle yield in the fractions of each individual run. The results in (**C**–**E**) are shown with the mean normalized intensity ± standard error. Statistical significance was tested using a two-way ANOVA with Fisher’s LSD post hoc test and denoted by an asterisk: * *p* < 0.05 (*n* = 6 for each sample type). Only significant differences due to the sample type are denoted.

**Figure 3 ijms-25-08130-f003:**
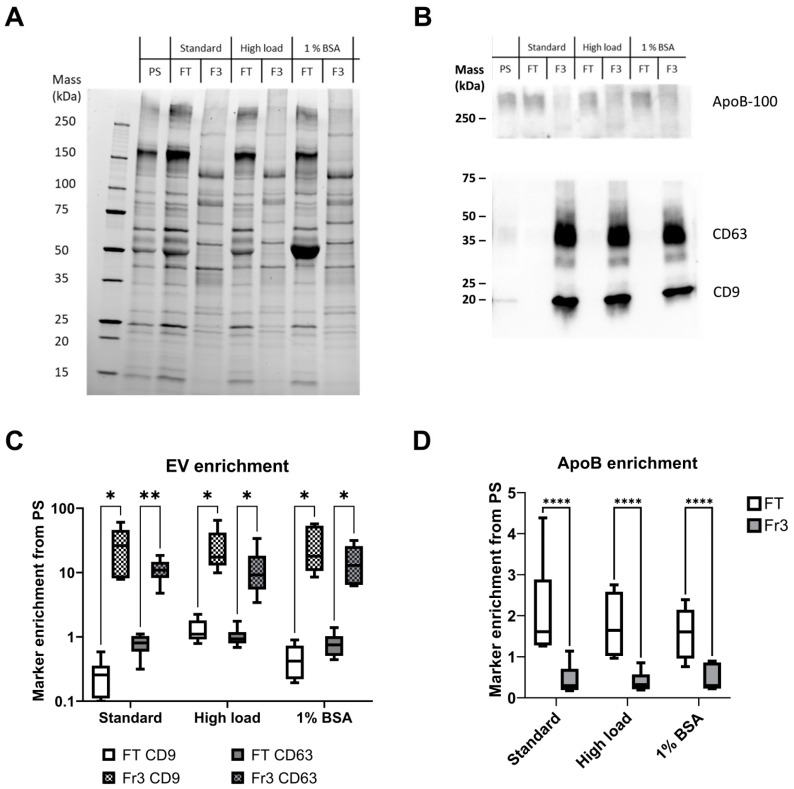
EVs are enriched from the platelet supernatant and separated from ApoB-positive lipoproteins. (**A**) Representative SDS-PAGE gel and (**B**) Western blot with EV markers CD9 and CD63 and ApoB of all the runs from single platelet supernatants (PSs). The peak fraction (Fr3) was used to represent the purified EVs from each sample. The gels were loaded using a normalized protein mass of 5 µg per well. (**C**) Assessment of EV markers CD9 and CD63 and (**D**) ApoB intensities from the Western blot assays in flowthrough (FT) versus Fr3. The results are shown as box and whisker plots, with marker intensities normalized against the PS. Statistical significance was tested using a two-way ANOVA with Fisher’s LSD post hoc test and denoted by an asterisk: * *p* < 0.05 or ** *p* < 0.01 or **** *p* < 0.0001 (*n* = 6 for each sample type). Only significant differences due to the fraction type are denoted; no significant differences were found due to sample type.

**Figure 4 ijms-25-08130-f004:**
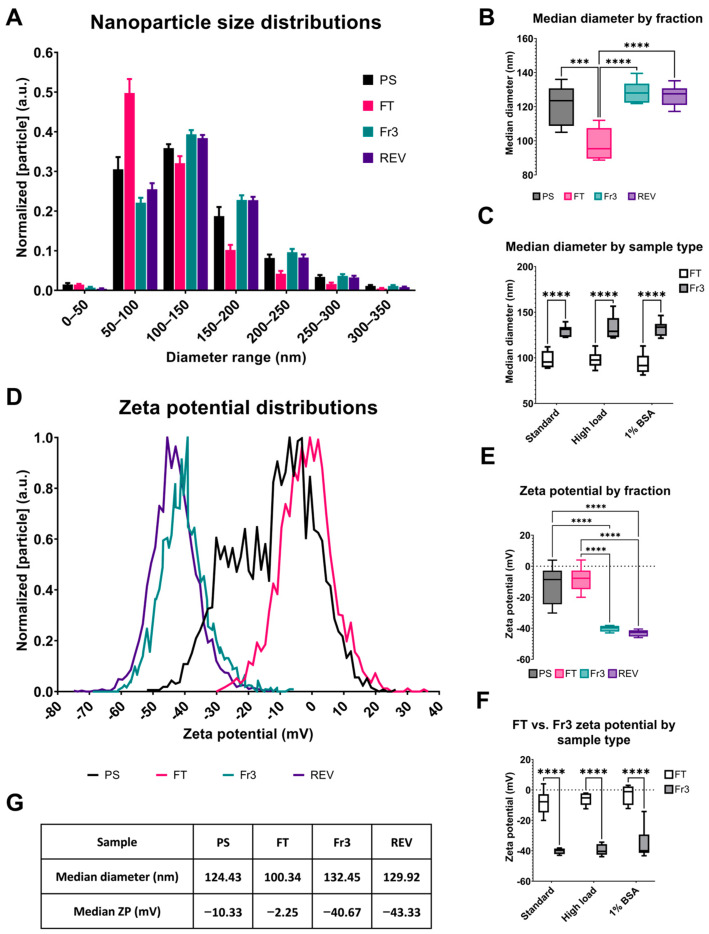
Flowthrough and peak fraction of the anion exchange chromatography contain nanoparticles with differing characteristics. The different sample types, platelet supernatants (PSs), flowthrough (FT), peak fractions (Fr3), and reference EVs (REV) were analyzed with Nanoparticle Tracking Analysis to determine their size and zeta potential distributions. (**A**) Mean size distributions of nanoparticles from the standard sample type. (**B**) Comparison of median particle diameters of standard sample fractions, PSs, and REVs and (**C**) between FT and Fr3 in different sample types. (**D**) Mean zeta potential distributions derived from standard sample fractions, PS, and REVs. (**E**) Comparison of median zeta potentials of standard sample fractions, PSs, and REVs and (**F**) between FT and Fr3 in different sample types. (**G**) Collected median diameters and zeta potentials of standard sample fractions, PSs, and REVs. Statistical significance was tested using one- (**B**,**E**) or two-way ANOVAs (**C**,**F**) with Fisher’s LSD post hoc test, denoted by an asterisk: *** *p* < 0.001 or **** *p* < 0.0001 (*n* = 6 for each sample type).

**Figure 5 ijms-25-08130-f005:**
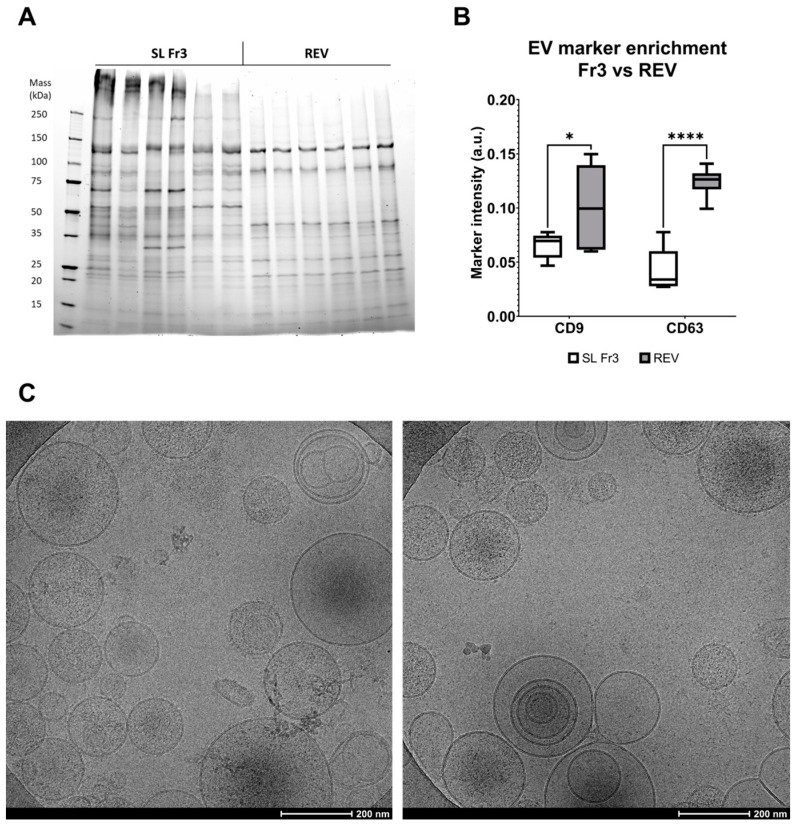
Anion exchange-purified EV samples contain more non-EV proteins compared to reference EVs. (**A**) SDS-PAGE gel and (**B**) Western blotting with normalized protein load (5 µg) were used to compare the overall protein composition and enrichment of EV markers between fraction 3 (Fr3) from the standard run (SL Fr3) and reference EVs (REV). The four first REV samples were prepared with a preconcentration step and the last two with direct injection into the SEC column, as described in the Section 4. A lower total band number in SDS-PAGE and Western blotting assessment for the enrichment of EV markers CD63 and CD9 showed that the markers were enriched significantly more in the REV samples, indicating higher EV purity. (**C**) Representative CryoTEM micrographs from two REV samples (#5 and 6 in the SDS-PAGE and Western blots of (**A**) and (**B**). Small lipid vesicles of varying sizes, shapes, and levels of lamellarity can be seen, indicating the presence of EVs. Statistical significance in (**B**) was tested using a one-way ANOVA with Fisher’s LSD post hoc test, denoted by an asterisk: * *p* < 0.05 or **** *p* < 0.0001 (*n* = 6 for each sample type).

**Figure 6 ijms-25-08130-f006:**
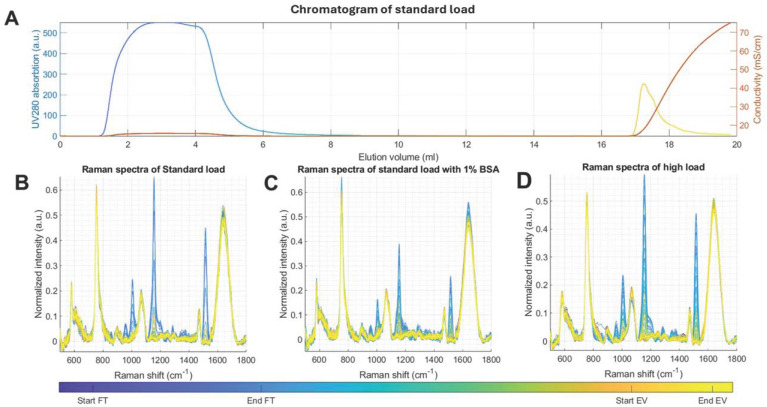
Inline Raman spectra collected during the purification of platelet EVs using anion-exchange chromatography. (**A**) Chromatogram with color-coded UV absorbance profile and conductivity using standard loading sample. (**B**) Inline Raman spectra of the chromatography fractions of standard, (**C**) high load, and (**D**) BSA sample types, respectively. Spectra and UV absorption profiles are colored according to the time and phase, as shown in the color bars at the bottom of the figure indicating the beginning and end of the process.

**Figure 7 ijms-25-08130-f007:**
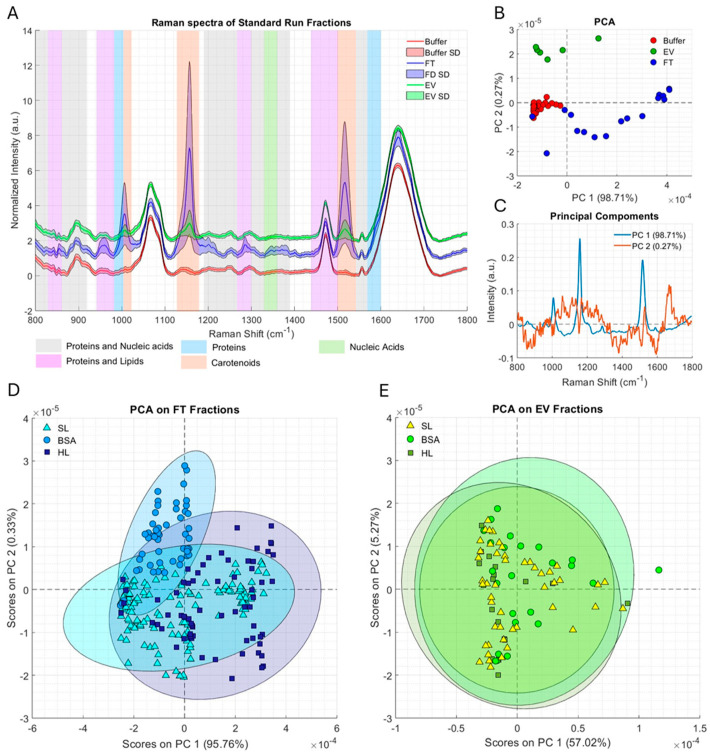
Raman spectra and principal component analysis on the chromatography fractions. (**A**) Raman spectra of the buffer (red), flowthrough (FT, blue), and eluted EV (green) fractions ± 1SD. The spectral alignments for different biomolecules are based on [26,27,28]. (**B**) PCA analysis; each dot represents a Raman spectrum collected during the run: buffer (red), FT (blue), and EVs (green), which all cluster in different areas. The partial overlap of FT and buffer is due to the low concentration of biological material in the FT. (**C**) The first and second principal component loading vectors, calculated from the full spectral data set measured during standard load (SL) run. (**D**) PCA of the FT fractions was collected using the three different sample types: SL (light blue triangle), high load (HL, dark blue square), and BSA (blue circle). (**E**) PCA of the EV fraction collected in the different sample types, SL (yellow triangle), HL (dark green square), and BSA (light green circle). Overlapping clusters indicate similar Raman spectra and, hence, similar biochemical composition.

**Figure 8 ijms-25-08130-f008:**
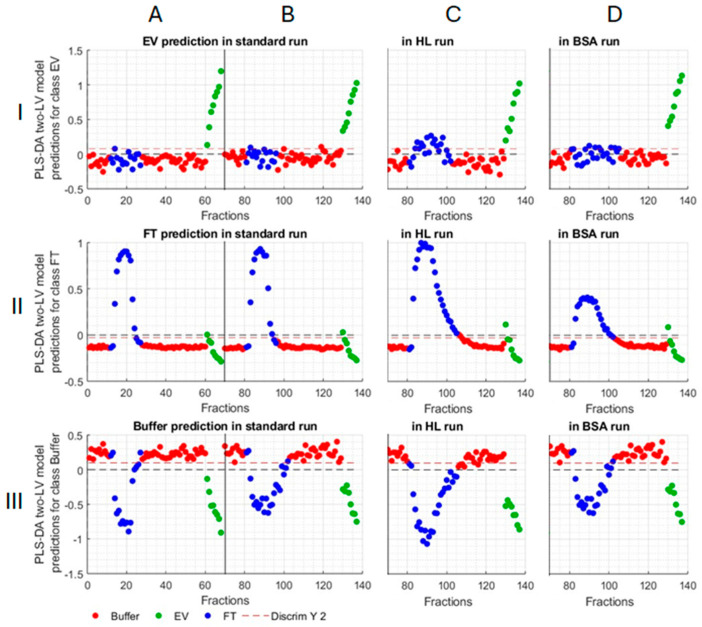
PLS-DA predictions of fraction types for standard, high load, and BSA sample types. (**A**) A standard sample run was always used as a training set; (**B**) 2nd standard run, (**C**) high load (HL), and (**D**) BSA, respectively, as the test sets. Training and test sets are divided by a vertical gray line, while the predictor is shown as a horizontal red dashed line. Each dot represents one Raman spectrum measured during the chromatography runs and divided based on the chromatography phases: buffer (red), FT (blue), and peak EV fraction (green). Dots above the predictor for each phase [(I) EV, (II) FT, or (III) buffer] are identified belonging to that phase.

**Table 1 ijms-25-08130-t001:** Identified Raman peaks and their assignments, based on [26,27,28].

Peak Region (cm^−1^)	Group/Bond	Assignment
750–870	Tyr, Trp	Protein
850–900	C-O-O	Lipid
810–900	DNA backbone	Nucleic acid
1000	Phe	Protein
1000	CH_3_	Carotenoid
900–1050	C-H	Lipid
1060–1180	C-C	Lipid/carotenoid
1250–1280	=CH	Lipid
1230–1270	Amide III	Protein
1250	CMP	Nucleic acid
1300	CH2	Lipid
1450	CH2-CH3	Protein/lipid
1520	Conjugated C=C-bonds	Carotenoid
1580	GMP, dTMP	Nucleic acid
1610–1620	Tyr, Trp, Phe	Protein
1650–1670	Amide I	Protein

**Table 2 ijms-25-08130-t002:** Sensitivity (the ability of the test to identify true positives, i.e., positive cases that were correctly identified) and specificity (the ability of the test to identify true negatives, i.e., negative cases that were correctly identified) ± 1SD of the 3 PLS-DA model.

Runs	Sensitivity	Specificity
Standard load		
Buffer	0.953 ± 0.002	0.896 ± 0.021
FT	0.875 ± 0.125	0.939 ± 0.041
EV	0.8435 ± 0.0315	0.983 ± 0.017
High load		
Buffer	0.752 ± 0.122	0.913 ± 0.004
FT	0.897 ± 0.022	0.847 ± 0.082
EV	0.937 ± 0.062	0.898 ± 0.034
1% BSA		
Buffer	0.860 ± 0.139	0.8375 ± 0.037
FT	0.869 ± 0.005	0.910 ± 0.067
EV	0.875 ± 0.002	0.932 ± 0.068

**Table 3 ijms-25-08130-t003:** Anion-exchange running protocol.

Step	Starting Point (mL)	Total Volume (mL)	A (%)	B (%)	Fractionation
Sample injection and wash	0	15	100	0	One 15 mL fraction (flowthrough, FT)
Elution	15	5	50	50	4 × 1 mL fractions, starting after 1 mL

## Data Availability

The original contributions presented in the study are included in the article, further inquiries can be directed to the corresponding author.

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
