# Peer review of "Inline Raman Spectroscopy Provides Versatile Molecular Monitoring for Platelet Extracellular Vesicle Purification with Anion-Exchange Chromatography"

_ijms, 2024, doi:10.3390/ijms25158130_

Round 1

Reviewer 1 Report

Comments and Suggestions for Authors

The manuscript by Saari et al. describes the use of inline Raman spectroscopy (RS) for assessing the purification of platelet extracellular vesicles (EVs) via monolith anion exchange chromatography. The objective of the manuscript is clear, well presented, and demonstrates that this detection system can be utilized for application development in this field. In my opinion, this work would interest the readers of IJMS. I have several comments for the authors to consider:

1. Figure 7B shows a PCA where PC1 explained 98.7% of the initial variance, and PC2 explained 0.27% of the variance. Similarly, Figure 7D shows PC2 explaining 0.3% of the initial variance. Did the authors consider alternative normalization and/or scaling strategies or using selected spectral ranges?

2. PLS-DA: Please describe the approach used for the training set selection. Did the authors include the SL set in both the training and test sets? Additionally, please explain the method used for selecting the number of latent variables (LVs).

3. The authors used Venetian blinds for PLS-DA CV. Did the authors consider using other types of CV (e.g., block-CV, CORRS-CV) to avoid including closely collected or consecutive spectra in both the training and test subsets during model development, or was it not required?

4. Did the model include the entire spectral region or an spectral subset? The authors might consider analyzing the loadings to identify the spectral regions associated with class differences.

- Please define the "true positives" in this context (Table 1).

- Why was EMSC used for spectral pre-processing in PLS-DA?

Author Response

Reviewer 1 response letter

We would like to thank the reviewer for their time and expert comments on our manuscript and encouraging words. 

The manuscript by Saari et al. describes the use of inline Raman spectroscopy (RS) for assessing the purification of platelet extracellular vesicles (EVs) via monolith anion exchange chromatography. The objective of the manuscript is clear, well presented, and demonstrates that this detection system can be utilized for application development in this field. In my opinion, this work would interest the readers of IJMS. I have several comments for the authors to consider:

  1. Figure 7B shows a PCA where PC1 explained 98.7% of the initial variance, and PC2 explained 0.27% of the variance. Similarly, Figure 7D shows PC2 explaining 0.3% of the initial variance. Did the authors consider alternative normalization and/or scaling strategies or using selected spectral ranges?

Answer: Yes, the spectral range used in PCA and PLS-DA was 800-1800 cm-1; leaving out the part of the spectra affected by the sapphire window peaks (400-800 cm-1) improved the result since the interference of the sapphire is removed.

  1. PLS-DA: Please describe the approach used for the training set selection. Did the authors include the SL set in both the training and test sets? Additionally, please explain the method used for selecting the number of latent variables (LVs).

Answer: Yes, SL was used different SL run for training and test. LV were chosen based on calibration and validation classification error average; CV classification error average was at its minimal when 2LV are used, see figure attached:

  1. The authors used Venetian blinds for PLS-DA CV. Did the authors consider using other types of CV (e.g., block-CV, CORRS-CV) to avoid including closely collected or consecutive spectra in both the training and test subsets during model development, or was it not required?

Answer: Yes, we did consider using different cross validation methods: however, in our opinion, venetian blind was suitable for the size and distribution of our datasets: in this CV method it is possible to set interval for data spit in a way that few spectra for each fractions (FT,EV and Buffer) are used as CV.

  1. Did the model include the entire spectral region or an spectral subset? The authors might consider analyzing the loadings to identify the spectral regions associated with class differences.

Answer: Yes, we tried also to “break down” the spectra in subset, however, best results could be achieved by using the interval 800-1800 cm-1 which leaves out the sapphire window interference as well as the “silent region” of the spectrum.

- Please define the "true positives" in this context (Table 1).

Answer: We define “true positives” as positive cases that were correctly identified. This has now been clarified in the Table legend (now “Table 2”).

- Why was EMSC used for spectral pre-processing in PLS-DA?

Answer: The Extended multiplicative scatter correction (EMSC) attempts to remove additive and multiplicative scattering effects in spectra. This helps to compensate the different “slopes” of the spectra (the flowthrough fractions have higher background since there is more material) making it more reliable than other baseline correction method. Once EMSC is applied, all the spectra had the same slope and there was no distortion when mean centering is applied.

Reviewer 2 Report

Comments and Suggestions for Authors

The authors reposted on using Ramn spectroscopy for monitoring the purity EVs. Although this concept is timely, there are a few issues that need to be addressed before suggesting for publication.

1-      The overall grammar and syntax of this manuscript would benefit from proof reading.

2-      It is not reflected on the title of this manuscript that anion exchange chromatography is used in junction with Raman.

3-      The authors mentioned in the abstract “Our results indicate a good separation of impurities from the EVs during …” could they quantify how good the proposed methods is compared to other available methods?

4-      The introduction could be further strengthened by adding other literature using Ramna for detection of Exosomes (e.g. doi.org/10.1002/anbr.202300055)

5-      Could the authors comment on how accurately they evaluated the intensity of CD9 and CD63? Is it done based on the intensity of WB, if so, this is not an accurate means of quantifying these markers.

6-      Why are the Zeta potential curves serrated?

7-      Could the authors label the Raman peaks in Fig.6 and 7

8-      In Figure.7 panel D and E, the PCA doesn’t show a very clear separation between the groups. Could the author comment on this and how this result has been reliably used to distinguish between the EVs?

Comments on the Quality of English Language

Moderate editing of English language required

Author Response

Reviewer 2 responses to comments.

We would like to thank the reviewer for their time and expert comments on our manuscript. 

The authors reposted on using Ramn spectroscopy for monitoring the purity EVs. Although this concept is timely, there are a few issues that need to be addressed before suggesting for publication.

1-      The overall grammar and syntax of this manuscript would benefit from proof reading.

      Answer: We agree that the language of the manuscript could have been improved for better clarity and readability and have now gone through the text and made extensive revisions to address this. We hope that the language now meets the high standards that are expected.

2-      It is not reflected on the title of this manuscript that anion exchange chromatography is used in junction with Raman.

      Answer: Indeed, the use of anion exchange chromatography was not introduced in the title, though it was an important aspect of our study. We have now changed the title accordingly to “Inline Raman spectroscopy provides versatile molecular monitoring for platelet extracellular vesicle purification with anion exchange chromatography”.

3-      The authors mentioned in the abstract “Our results indicate a good separation of impurities from the EVs during …” could they quantify how good the proposed methods is compared to other available methods?

Answer: The comparison of purification methods for EV isolation is indeed topical, especially regarding such emerging new methods as ion exchange chromatography. Since we did not perform a systematic comparison of other methods to the monolith anion exchange chromatography employed here, we feel that it is out of the scope of our study to compare our EVs to other methods regarding purity. In our study we did also use purified platelet EVs as a reference (reference EVs in the text) that were produced with ultracentrifugation and size exclusion chromatography and found those EVs to be of higher purity.

4-      The introduction could be further strengthened by adding other literature using Ramna for detection of Exosomes (e.g. doi.org/10.1002/anbr.202300055)

      Answer: We agree that the Raman aspect of the introduction could have used more references. Given the large number of studies published regarding the subject, we have opted to adding two recent review papers about the subject (line 107).

 In the article suggested by the reviewer, SERS  based metho is used to discriminate between types of EV, we don´t think it fits with quality control/process monitoring approach we are suggesting

5-      Could the authors comment on how accurately they evaluated the intensity of CD9 and CD63? Is it done based on the intensity of WB, if so, this is not an accurate means of quantifying these markers.

      Answer: We agree that western blotting is not an accurate quantitative method, and this has now been highlighted in the text (lines 161-162). As a semi-quantitative method, trends of the protein expression can still be observed that are in line with our other results.

6-      Why are the Zeta potential curves serrated?

      Answer: This is most likely due to the instrument used for Zeta potential assessment (ZetaView). The instrument is basically an ultramicroscope that follows the motion of nanoparticles in an electric field with a video analysis. While the advantage of this approach is the analysis of single particles instead of bulk analysis, the number of particles per analysis is limited to about 400 per sample, which was in our case repeated with six independent samples. For this reason, the method is still somewhat sensitive to outliers or high heterogeneity, which cause the observed serration in the final result, instead of smooth curves. It is not clear at this point, if this is an artefact result or more accurate than bulk analysis type methods.

7-      Could the authors label the Raman peaks in Fig.6 and 7

      Answer: The peaks are now identified in new table (Table 1, line 333).     

8-      In Figure.7 panel D and E, the PCA doesn’t show a very clear separation between the groups. Could the author comment on this and how this result has been reliably used to distinguish between the EVs?

      Answer: In figure 7D, the flowthrough Raman spectra of different sample types are compared. As standard load and high load samples are the same in composition, no separation is to be expected and this is observed in the analysis. The BSA sample however forms its own cluster that can be mostly separated from the others. This is due to the added BSA, which causes a significant difference in the composition of the sample compared to the others, which is reflected into the Raman spectra. The exception to this are the points measured at the low concentration positions of the flowthrough (“tail regions”). At these points the collected spectra are not intense enough to establish any clear differences from the other sample type tail regions.

      In figure 7E the situation is similar in the elution fractions. However, this time not even the BSA samples in peak position could be differentiated from the other sample types, suggesting that there were no significant molecular differences between any of these groups, i.e. the eluate was the same regardless of the sample type.

      To conclude, we find that the lack of differentiation besides the BSA flowthrough is the main result. This was in line with our downstream analyses (fig. 2 – 4) that also found the elution fractions to be near identical in protein- and nanoparticle composition, and the added BSA was indeed confirmed to being present in the flowthrough fraction. These are also our conclusions described in the results and discussion sections.

Reviewer 3 Report

Comments and Suggestions for Authors

The manuscript describes the use of Raman spectroscopy for inline monitoring of the process of platelet extracellular vesicle purification. This is a potentially important application of the technology, and therefore the study is of significance. However, there are several issue which should be considered to improve the quality of the presentation;

(i) EVs can be up to tens of micrometers in size, and therefore it is not correct generally characterise them as nanoparticles, which in many official definitions are particles of at least one dimension <100nm. 

(ii) The context of the use of the term "novel" to describe EVs is not clear. Although we may have only discovered them relatively recently, presumably they have been present since humans, and before.

(iii) The first 9-10 lines of the Abstract are more of an Introduction, and only the latter half appropriate for an Abstract. The Abstract should describe what was done and how, what was observed, and what was concluded.

(iv) Many statements requires supporting reference(s) e.g.

"EVs function as carriers of biomolecular information, including proteins, lipids, nucleic acids and metabolites that affect their recipient cells in various ways."

"....many research groups and commercial organizations are developing EVs for different therapeutic applications, as carriers of drugs or as biomarkers for diagnostics."

"Large-scale isolation methods for EVs are being developed as well to help answer the demand for high quality and quantity of EVs that are needed for these novel applications."

"Various types of inline detectors for methods using a liquid flow, such as liquid chromatography, are available and have been applied with EV purification."

"Turbidity is usually more distinguishable at longer wavelengths, where biological samples typically do not have high absorbance (> 400 nm), and therefore do not cause high back ground."

"Light scattering is one of the most used approaches for analysing EVs overall, as it can be used to detect and measure EVs with good sensitivity."

"Raman spectroscopy has exceptional potential as a method for process monitoring of EVs and other biological products with a complex chemical structure."

(iv) As the Introduction progresses, and into the presentation of Results, the manuscript becomes increasingly difficult to follow because of numerous acronyms and parameters which have not been explained. All of these need to be clearly explained:

A280 nm, A260 nm, A260/A280, SEC, V/LDL particles, EVs from the PS, loss in the FT, 1 % BSA 132 (SB), end of Fr2,with most of the peak collected in Fr3.

(v) There are several very imprecise statements, e.g. "Besides absorbance, the attenuation of light, which is the quantity measured by the UV-detectors...." UV detectors measure UV light....

(vi) "assessing EV purity by our group and others." The article has 4 authors from 4 different affiliations, what does "our group" mean in this context?

(vii) Figure 1 should be described in the text.

(viii) Figure 2a - Absorbance is the log of a ratio, and has no units, not (a.u)

(ix) What is plotted in Figure 3(a)? If measured by DLS, this should be particle number (rather than Intensity).

(x) Similarly, the authors should clarify what is plotted in Figure 3(b)

(xi) In describing the Raman measurements, detains should be provided of laser power, spot size, etc.

(xii) The authors should explain, on what basis were 2 latent variables chosen fro the PCA-LDA

(xiii) "Among the identified Raman peaks, we found that platelet EVs contain also carotenoids," It is not quite clear how this conclusion was made - the authors should better describe.

Author Response

Reviewer 3 response letter

We would like to thank the reviewer for their time and expert comments on our manuscript.

The manuscript describes the use of Raman spectroscopy for inline monitoring of the process of platelet extracellular vesicle purification. This is a potentially important application of the technology, and therefore the study is of significance. However, there are several issue which should be considered to improve the quality of the presentation;

(i) EVs can be up to tens of micrometers in size, and therefore it is not correct generally characterise them as nanoparticles, which in many official definitions are particles of at least one dimension <100nm. 

Answer: We agree with the reviewer that our description of EV size range was not accurate as larger EVs also have been described. We have now included this in our introduction (line 33).

(ii) The context of the use of the term "novel" to describe EVs is not clear. Although we may have only discovered them relatively recently, presumably they have been present since humans, and before.

Answer: Indeed EVs may be novel to us as researchers, but as a natural phenomenon they are not necessarily novel at all. To clarify this, we have decided not to describe EVs as novel in the text.

(iii) The first 9-10 lines of the Abstract are more of an Introduction, and only the latter half appropriate for an Abstract. The Abstract should describe what was done and how, what was observed, and what was concluded.

Answer: We agree that the abstract was not properly structured, being disproportionately introductory. We have now modified the abstract to better correspond to the recommended format.

(iv) Many statements requires supporting reference(s) e.g.

Answer: We have now added references to the recommended statements and clarified them as being more concise. Changes are denoted after each instance mentioned by the reviewer.

"EVs function as carriers of biomolecular information, including proteins, lipids, nucleic acids and metabolites that affect their recipient cells in various ways."

Added reference, line 37.

"....many research groups and commercial organizations are developing EVs for different therapeutic applications, as carriers of drugs or as biomarkers for diagnostics."

This sentence was removed as being imprecise.

"Large-scale isolation methods for EVs are being developed as well to help answer the demand for high quality and quantity of EVs that are needed for these novel applications."

This is now explained in the subsequent sentence, with added reference, line 43.

"Various types of inline detectors for methods using a liquid flow, such as liquid chromatography, are available and have been applied with EV purification."

This sentence was removed as no reference could be found.

"Turbidity is usually more distinguishable at longer wavelengths, where biological samples typically do not have high absorbance (> 400 nm), and therefore do not cause high back ground."

Added reference, line 70.

"Light scattering is one of the most used approaches for analysing EVs overall, as it can be used to detect and measure EVs with good sensitivity."

Added NTA as an example with reference, line 72.

"Raman spectroscopy has exceptional potential as a method for process monitoring of EVs and other biological products with a complex chemical structure."

Added two references, line 107.

(iv) As the Introduction progresses, and into the presentation of Results, the manuscript becomes increasingly difficult to follow because of numerous acronyms and parameters which have not been explained. All of these need to be clearly explained:

A280 nm, A260 nm, A260/A280, SEC, V/LDL particles, EVs from the PS, loss in the FT, 1 % BSA 132 (SB), end of Fr2,with most of the peak collected in Fr3.

Answer: These acronyms and others were indeed not clarified to the reader and have now been explained in the text.

(v) There are several very imprecise statements, e.g. "Besides absorbance, the attenuation of light, which is the quantity measured by the UV-detectors...." UV detectors measure UV light....

Answer: We agree that the text contained several imprecise statements. We have now gone through the whole manuscript (as suggested by another reviewer as well) to improve the language to being more precise and readable.

(vi) "assessing EV purity by our group and others." The article has 4 authors from 4 different affiliations, what does "our group" mean in this context?

Answer: We agree that while the authors were mostly the same, this statement was unnecessary and misleading. This has now been removed.

(vii) Figure 1 should be described in the text.

Answer: Figure 1 is now described in the text (lines 94 – 100).

(viii) Figure 2a - Absorbance is the log of a ratio, and has no units, not (a.u)

Answer: This has now been corrected into figure 2. Absorbance also can be described as “absorbance unit” (AU or mAU), which is common in chromatography, and we also use it here.

(ix) What is plotted in Figure 3(a)? If measured by DLS, this should be particle number (rather than Intensity).

Answer: Assuming the reviewer is referring to Figure 4A, it is measured with NTA, not DLS. Nevertheless, the unit is still normalized particle concentration, as is depicted in the figure.

(x) Similarly, the authors should clarify what is plotted in Figure 3(b)

Answer: We are not certain, which figure the reviewer is referring to. Figure 4B seems clear to us. Figure 4D has the same unit as Figure 4A, though it is presenting the Zeta potentials of the particles.

(xi) In describing the Raman measurements, detains should be provided of laser power, spot size, etc.

Answer: The missing details of the Raman measurements have now been added (line 594 – 595).

(xii) The authors should explain, on what basis were 2 latent variables chosen fro the PCA-LDA

Answer: SL was used different SL run for training and test. LV were chosen based on calibration and validation classification error average; CV classification error average was at its minimal when 2LV are used, see figure attached:

(xiii) "Among the identified Raman peaks, we found that platelet EVs contain also carotenoids," It is not quite clear how this conclusion was made - the authors should better describe.

Answer: Carotenoids resonate with 532 laser and the resulting spectra shows three clear peak at 1006 1154 and 1510 cm-1 [https://www.nature.com/articles/s41598-020-64737-3], these peaks are clearly present in EV fractions and they are disproportionately strong. This explanation has been added into the text (lines 449 – 450).

Round 2

Reviewer 1 Report

Comments and Suggestions for Authors

I appreciate the authors' response to the initial comments. Regarding their feedback:

1.           Figure 7B shows a PCA where PC1 explained 98.7% of the initial variance, and PC2 explained 0.27% of the variance. Similarly, Figure 7D shows PC2 explaining 0.3% of the initial variance. Did the authors consider alternative normalization and/or scaling strategies or using selected spectral ranges? Answer: Yes, the spectral range used in PCA and PLS-DA was 800-1800 cm-1; leaving out the part of the spectra affected by the sapphire window peaks (400-800 cm-1) improved the result since the interference of the sapphire is removed.

* The response provided does not address the initial comment regarding the 0.27% variance found in the second PC. This is very unusual, and I would recommend discussing this value in the manuscript.

4.           Did the model include the entire spectral region or an spectral subset? The authors might consider analyzing the loadings to identify the spectral regions associated with class differences. Answer: Yes, we tried also to “break down” the spectra in subset, however, best results could be achieved by using the interval 800-1800 cm-1 which leaves out the sapphire window interference as well as the “silent region” of the spectrum

* I would strongly encourage the authors to describe the main spectral regions driving the separation. While the exclusion of the sapphire window region is justified due to interference, it is also important to discuss the specific discriminant spectral regions, within the 800-1800 cm-1 range. Highlighting these regions might enhance the robustness of the analysis and also provide valuable insights into the underlying properties responsible for the observed class differences. I find that this additional detail might significantly improve the discussion and interpretability of the results.

Author Response

  1. Figure 7B shows a PCA where PC1 explained 98.7% of the initial variance, and PC2 explained 0.27% of the variance. Similarly, Figure 7D shows PC2 explaining 0.3% of the initial variance. Did the authors consider alternative normalization and/or scaling strategies or using selected spectral ranges? Answer: Yes, the spectral range used in PCA and PLS-DA was 800-1800 cm-1; leaving out the part of the spectra affected by the sapphire window peaks (400-800 cm-1) improved the result since the interference of the sapphire is removed.

* The response provided does not address the initial comment regarding the 0.27% variance found in the second PC. This is very unusual, and I would recommend discussing this value in the manuscript.

-We apologize for missing part of the original comment in our response. PC1 explains such a high portion of variance because it is based on the strongest peaks in the spectra, related to carotenoids. The peaks related to PC2 are not as clear, but a few can be identified related to lipids and proteins (notably the C-H and Amide I peaks). The difference in PC1 carotenoid peak intensities was clear between flowthrough and elution fractions, which is likely due to the separation of lipoproteins that carry carotenoids into the flowthrough (as discussed in the discussion-section). As the peaks related to PC2 are not as specific to any particularly separated molecular entity, it cannot be as strong as an explanatory factor as PC1. This point is now addressed in the manuscript, lines 352 – 356 and 460 – 468. Alternative strategies for spectral analysis could of course be employed, producing different results, but in this case we found our approach as the most straightforward choice.

  1. Did the model include the entire spectral region or an spectral subset? The authors might consider analyzing the loadings to identify the spectral regions associated with class differences. Answer: Yes, we tried also to “break down” the spectra in subset, however, best results could be achieved by using the interval 800-1800 cm-1 which leaves out the sapphire window interference as well as the “silent region” of the spectrum

* I would strongly encourage the authors to describe the main spectral regions driving the separation. While the exclusion of the sapphire window region is justified due to interference, it is also important to discuss the specific discriminant spectral regions, within the 800-1800 cm-1 range. Highlighting these regions might enhance the robustness of the analysis and also provide valuable insights into the underlying properties responsible for the observed class differences. I find that this additional detail might significantly improve the discussion and interpretability of the results.

-We agree that describing the most notable spectral regions is highly beneficial for the analysis of our results. We have now included description of these regions in the text in lines 315 – 316 and 352 – 356 also as in our answer to the above comment #1. Added Table 1 also provides a list of peaks and their alignments for their easier identification.